# The Historical Complexity of Tree Height Growth Dynamic Associated with Climate Change in Western North America

Yassine Messaoud [1,*], Anya Reid [2], Nadezhda M. Tchebakova [3], Jack A. Goldman [4] and Annika Hofgaard [5]

1 Faculty of Natural Resources Management, Lakehead University, 955 Oliver Road, Thunder Bay, ON P7B 5E1, Canada

2 Forest Inventory, BC Government Directory, P.O. Box 9512, Stn Prov Govt, Victoria, BC V8W 9C2, Canada; anya.reid@gov.bc.ca

3 Sukachev Institute of Forest, Siberian Branch, Russian Academy of Sciences, Krasnoyarsk 660036, Russia; ncheby@ksc.krasn.ru

4 Institute of Forestry and Conservation, John H. Daniels Faculty of Architecture, Landscape and Design, University of Toronto, Toronto, ON M5S 1A1, Canada; jack.goldman@mail.utoronto.ca

5 Norwegian Institute for Nature Research, NO-7485 Trondheim, Norway; annika.hofgaard@nina.no

* Correspondence: ymessaou@lakeheadu.ca

**Abstract:** The effect of climate on tree growth has received increased interest in the context of climate change. However, most studies have been limited geographically and with respect to species. Here, sixteen tree species of western North America were used to investigate the response of trees to climate change. Forest inventory data from 36,944 stands established between 1600 and 1968 throughout western North America were summarized. The height growth (top height at a breast-height age of 50 years) of healthy dominant and co-dominant trees was related to annual and summer temperatures, the annual and summer Palmer Drought Severity Indexes (PDSIs), and the tree establishment date (ED). Climate-induced height growth patterns were then tested to determine links to the spatial environment (geographic locations and soil properties), the species' range (coastal, interior, or both), and traits (shade tolerance and leaf form). Analysis was performed using a linear mixed model (total species) and a general linear model (species scale). Climate change was globally beneficial, except for Alaska yellow-cedar (*Chamaecyparis nootkatensis* (D. Don) Spach), and growth patterns were magnified for coastal-ranged, high-shade-tolerant, and broadleaf species, and mostly at the northernmost extents of these species' ranges. Nevertheless, growth patterns were more complex with respect to soil properties. A growth decline for some species was observed at higher latitudes and elevations and was possibly related to increased cloudiness, precipitation, or drought (in interior areas). These results highlight the spatio-temporal complexity of the growth response to recent global climate change.

**Keywords:** height growth; site index; global climate change; species range; species characteristics; species ecological amplitude; geographic locations; western North America

## 1. Introduction

Understanding recent climate change effects on tree growth is necessary to accurately forecast species and forest dynamics under future climate change scenarios [1]. Numerous studies highlight the effect of recent global temperature increases on tree growth around the world, especially at higher latitudes and elevations where temperature limits growth [2–6]. In many studies, tree growth is positively associated with increasing temperature as a result of global climate change [7–10]; however, growth decline is observed in water-stressed environments [11,12], with tree dieback becoming more obvious in areas with severe drought [13].

Understanding how the magnitude of the growth response to climate may differ based on species' growth habits and autecology is necessary to improve predictive modeling.

In optimal environmental conditions, photosynthesis increases with increasing temperatures [14]. Thus, faster-growing species (shade-intolerant) may more significantly increase their photosynthetic capacity, and consequently their growth, in response to increased temperatures compared to slower-growing species (shade-tolerant), which generally have lower light saturation points for photosynthesis [14]. In boreal forests, the faster-growing genera *Betula* and *Populus* tend to have greater responses to increasing temperatures than the slower-growing genera *Pinus* and *Picea* [15]. Similarly, Saxe et al. [16] reported that broadleaved species, such as trembling aspen (*Populus tremuloides* Michx.) and paper birch (*Betula papyrifera* Marsh.), exhibited a higher relative growth rate under higher temperatures than the conifer species tamarack (*Larix laricina* (Du Roi) K. Koch), black spruce (*Picea mariana* (Mill.) B. S. P.), and jack pine (*Pinus banksiana* (Lamb.)), suggesting different growth patterns with different leaf forms. However, this trend is not universal. Woodward [15] reported that in the same environmental conditions, Norway spruce (*Picea abies* (L.) Karst), a coniferous species, had a better relative growth rate and height increment than the deciduous species European beech (*Fagus sylvatica* L.). These opposite findings may be related to the difference in shade tolerance between the deciduous species; trembling aspen and paper birch are shade-intolerant, while European beech is very shade-tolerant. Therefore, growth habits may also play a big role in a tree's response to climate change [17]. A study on the effects of paleoenvironmental changes on the interspecific differences in life-history traits of tree species [18] found that climate change during the cold period following deglaciation favored species with a 'fast' life-history strategy (e.g., a high relative growth rate, a short life span, and shade intolerance), such as lodgepole pine (*Pinus contorta* Dougl. ex. Loud.). In contrast, the relative climatic stability of the last ca. 8000 years favored species that exhibit a 'slow' life-history strategy (e.g., a low relative growth rate, a long life span, and high shade tolerance), such as western hemlock (*Tsuga heterophylla* (Raf.) Sarg.).

In forest ecosystems, tree growth is limited by energy (growing season length, degree-days, or temperatures) in some regions and by soil properties (moisture and nutrient regimes) in other regions [19,20]. For instance, several studies reported a positive growth response to recently increased temperatures at higher latitudes or elevations where temperature is usually the limiting factor [21]. At lower latitudes or elevations, however, the increased water deficits associated with temperature increases can lead to declining tree growth [22–24]. Similar declines were also observed in the interior boreal forest of Alaska, in the prairies, and in the southern interior of British Columbia, with higher rates of tree mortality associated with a warmer climate [25–28].

The pattern of increased or decreased growth with recent climate change is globally still unclear. In western North America, similar growth patterns were reported regardless of the species [29], while other studies found species-specific growth responses to climate change [17,30–32]. These contradictory findings may be related to the scale of the study area and/or the studied species. Most studies are conducted at smaller scales (except Messaoud and Chen [30]) and on a few species (except Hararuk et al. [29]). Moreover, Miyamoto et al. [33] observed that, although tree growth for subalpine fir, lodgepole pine (*Pinus contorta* Dougl. ex. Loud.), and white spruce (*Picea glauca* (Moench) Voss) was strongly temperature-sensitive on a large scale, i.e., when examining all of western North America, the climate-change-related response growth patterns were nevertheless species-specific and site-specific; this corroborated with those from European tree species [23]. Thus, it is crucial to examine the historical tree growth for the full species' range to capture the whole pattern of historical tree growth response in order to increase our knowledge of future forest dynamics and species' distribution [34].

In addition, spring snowpack accumulation appears to have a contrasting effect; indeed, Peterson and Peterson [32] mentioned that spring snowpack accumulation negatively affected tree growth for subalpine fir (*Abies lasiocarpa* (Hook.) Nutt.), Engelmann spruce (*Picea engelmannii Parry ex Engelm.*), and subalpine larch (*Larix lyallii Parl.*) in areas with more limited temperatures, i.e., at high elevations of the Cascade Mountains of west-

ern North America. In contrast, the spring snowpack had a positive effect on tree growth in areas where water is more limited, i.e., in pine forests at low latitudes and elevations [35]. In addition, Lévesque et al. [34] pointed out the importance of soil moisture and nutrients on climate-change-induced tree growth patterns. Indeed, they found that high soil nutrient and water availability significantly promoted and magnified tree growth patterns for most of the dominant tree species in central Europe. In conclusion, they argued that many studies that related tree growth to climate variability did not include the effects of the soil properties.

Studies conducted mostly in the transition area between the forest and tundra zones show that other factors, for example, soil properties and disturbances (wind, fire, and grazing), may alter or mask the response to climate change [36,37]. Several studies used stem radial increments to reconstruct growth and correlate it with recent climate change [10,38]; however, in comparison to radial growth, height growth is more strongly dependent on climate and soil properties and less affected by stand density [39–42], making it a complementary method for examining the response of tree species to climate change.

This study examines the long-term effect of global climate change (temperatures and soil moisture availability) on the height growth of 16 tree species of western North America. We hypothesize that (1) similar to recent findings regarding tree radial growth, there is also a positive height growth response to global warming; (2) growth responses differ according to species' growth habits (fast vs. slow growth) and leaf form (needle leaves, deciduous needle leaves, and deciduous broadleaves); and (3) growth responses differ according to the spatial environment based on the resource limitation theory.

## 2. Materials and Methods

### 2.1. Study Area

The study area covers western Canada and the western U.S., ranging from $31°24'$ to $62°08'$ N latitude and $99°31'$ to $153°51'$ W longitude and at elevations from 5 to 3687 m above sea level (Table 1; Figure 1). This area of western North America contains one of the most diverse ecoregions in the world (Figure S1).

**Table 1.** Characteristics of the study plots showing ranges in latitude, longitude, elevation, establishment date (ED), and site index at age 50 (SI).

| Species | N | Latitude | Longitude (W) | Elevation (m) | ED | SI (m) |
|---|---|---|---|---|---|---|
| Alaska-cedar | 445 | $44°24'$–$58°50'$ | $121°30'$–$136°23'$ | 8–1781 | 1601–1940 | 2.78–29.5 |
| Amabilis fir | 791 | $42°56'$–$55°49'$ | $120°37'$–$130°21'$ | 1–1920 | 1604–1968 | 2.02–48.52 |
| Black cottonwood | 99 | $43°33'$–$61°19'$ | $109°05'$–$153°51'$ | 30–2048 | 1783–1965 | 8.83–41.60 |
| Douglas-fir | 13,987 | $31°24'$–$55°02'$ | $104°19'$–$127°26'$ | 5–3477 | 1600–1966 | 2.44–53.34 |
| Engelmann spruce | 3899 | $32°42'$–$56°59'$ | $104°56'$–$128°23'$ | 610–3687 | 1634–1964 | 2.85–32.24 |
| Grand fir | 1345 | $39°06'$–$50°14'$ | $113°47'$–$125°24'$ | 25–2103 | 1739–1966 | 6.10–59.89 |
| Lodgepole pine | 6784 | $34°07'$–$62°08'$ | $104°58'$–$136°14'$ | 11–3612 | 1604–1966 | 2.13–32.92 |
| Mountain hemlock | 1825 | $37°09'$–$61°06'$ | $114°36'$–$150°36'$ | 7–3150 | 1600–1960 | 2.01–27.01 |
| Ponderosa pine | 6838 | $31°47'$–$51°12'$ | $99°31'$–$124°03'$ | 91–3036 | 1625–1967 | 4.07–49.53 |
| Red alder | 135 | $40°45'$–$57°59'$ | $121°09'$–$135°28'$ | 5–1072 | 1781–1963 | 14.54–42.31 |
| Sitka spruce | 922 | $40°34'$–$61°07'$ | $121°47'$–$153°11'$ | 1–1255 | 1603–1962 | 2.31–57.91 |
| Subalpine fir | 5296 | $33°36'$–$61°12'$ | $105°06'$–$133°04'$ | 100–3627 | 1624–1967 | 1.82–40.39 |
| Western hemlock | 3479 | $37°55'$–$61°04'$ | $114°29'$–$148°04'$ | 1–2896 | 1601–1967 | 2.33–43.28 |
| Western larch | 1006 | $44°18'$–$50°50'$ | $112°51'$–$121°16'$ | 455–2286 | 1617–1964 | 7.2–34.14 |
| Western redcedar | 1424 | $43°35'$–$58°20'$ | $113°44'$–$135°06'$ | 1–1783 | 1601–1962 | 3.52–36.86 |
| Western white pine | 165 | $36°18'$–$52°13'$ | $113°50'$–$124°28'$ | 91–3018 | 1659–1963 | 5.57–36.58 |
| *Total* | *48,440* | *$31°24'$–$62°08'$* | *$99°31'$–$153°51'$* | *1–3687* | *1600–1968* | *1.82–59.89* |

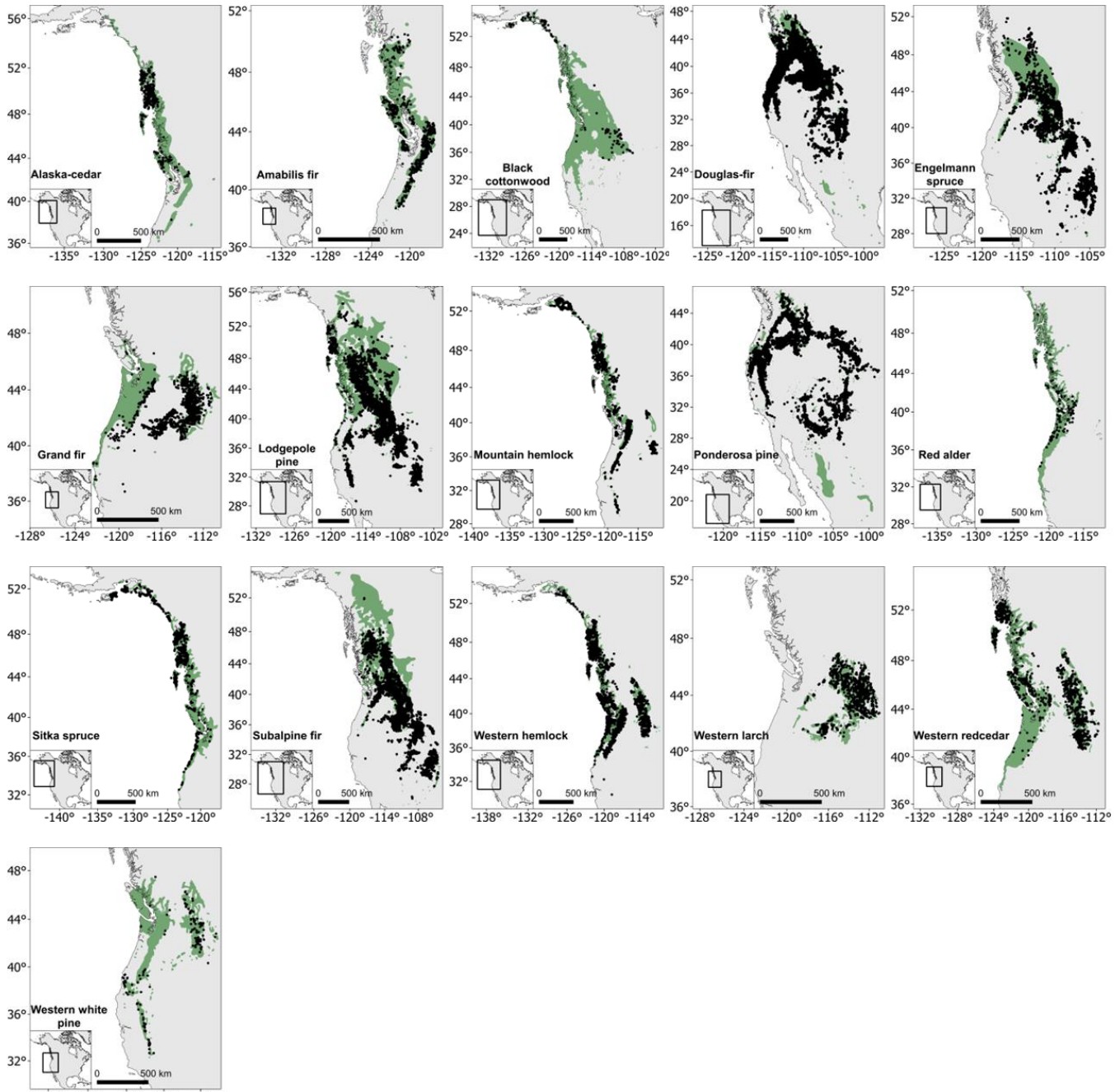

**Figure 1.** Distribution of sample plots of the sixteen study species in western North America. Each map also indicates species' geographic range (in green).

### 2.2. Sampling Design

The study uses forest inventory data provided by government agencies in Canada and the U.S. (FIA). For Canada, the dataset originated from the SIBEC project (Site Index Biogeoclimatic Ecosystem Classification). This project reflects the average growth potential of tree species in forested sites in British Columbia. Data from Yukon and Alberta originated from permanent sample plots. In total, 36,944 sample stands were selected, spanning from Alaska to southern California (Figure 1). The forest inventory database has been maintained since the 1930s. All selected stands were naturally established after a wildfire, were unmanaged, and without visible damage or disturbance. These stands were of variable ages, and for a given period of growth (the first 50 years) at breast height, stands experienced a wide range of historical growing conditions with respect to temperature

associated with climate change. Within stands, a plot size ranged from 0.01 to 0.5 ha in Canada and from 0.01 to 0.017 ha in the U.S. (Forest Inventory and Analysis national Program: https://www.fia.fs.fed.us/library/database-documentation/index.php, accessed on 2 April 2022). For each plot, information include the geographic location (latitude, longitude, and elevation), the stand condition (i.e., visible natural or anthropic disturbance, treatment, and plantation or natural establishment), the year of measurement (1937–2018), and the selected trees for the site index (top tree height, diameter at the breast height, breast-height age, total age, and site index).

### 2.3. Tree Height and Site Index

Within a plot, the dominant and co-dominant (largest trees) living trees for each species were aged by counting tree rings from an increment core sample extracted at the breast height of a tree, and the total diameter was measured at the root collar for each species. These trees were free from suppression above breast height, had no visible damage, and had a full crown. Only trees with total age ≥ 50 years were selected. Then, we averaged the height (measured with a clinometer or other approved instrument from the ground to the top of the tree) and age for each species within each plot. Site index (SI) is defined as the height of the dominant and co-dominant trees at the reference age, usually 50 years at breast height, and is a strong predictor of forest stand productivity. The SI was calculated using the top tree height of the dominant and co-dominant sample trees at a reference age of 50 at breast height; this was carried out for each tree of each species using the software Site Tools 4.1 Beta developed by the Forest Analysis and Inventory Branch, British Columbia Ministry of Forests (TIPSY v4.4: https://www2.gov.bc.ca/gov/content/industry/forestry/managing-our-forest-resources/forest-inventory/field-forms-and-software/software-download#tipsy, accessed on 2 April 2022). This modeling software is a growth and yield program allowing the calculation of the site index for each tree according to the height–age model equation developed for each species using the top height and total age of the sample tree. The establishment date was estimated by subtracting the tree age from the year of measurement. The establishment date of all species was site-specific but ranged between 1600 and 1968. Thus, we were able to figure out the evolution of the SI over time and highlight possible increases (higher SI) or declines (lower SI) within and between species.

### 2.4. Study Species

Sixteen tree species were analyzed, including conifers and broadleaves with different ecologies, shade tolerances, and geographic ranges (Figure 1; Table S1). The descriptions of the studied species were provided by Burns and Honkala in [43,44]. These species together account for more than 60 percent of the wide-ranging tall- and medium-sized tree species (taller than 9 m in height) in western North America [45].

The ranges of Alaska yellow-cedar (*Chamaecyparis nootkatensis* (D. Don) Spach), Amabilis fir (*Abies amabilis* Dougl. ex Loud. Forbes), Sitka spruce (*Picea sitchensis* (Bong.) Carr.), and red alder (*Alnus rubra* Bong.) are restricted to the Pacific coast and coastal mountains (i.e., the Cascades) (Figure 1). Conversely, Engelmann spruce (*Picea engelmannii* Parry ex Engelmann) and subalpine fir (*Abies lasiocarpa* Hook Nutt.) are located and associated with each other throughout the interior Rocky Mountains, mostly at high elevations. Western larch (*Larix occidentalis* Nutt) is also located in the interior; its range is restricted to the Rockies.

Six species have two distinct ranges: along the Pacific coast and the coastal mountains and throughout the Rocky Mountains. Douglas-fir (*Pseudotsuga menziesii* (Mirb.) Franco) ranges from central British Columbia to central California, but its interior range extends farther south to central Mexico. Although mountain hemlock (*Tsuga mertensiana* (Bong.) Carr.), western hemlock (*Tsuga heterophylla* Raf. Sarg.), and western redcedar (*Thuja plicata* Donn ex D. Don Spach) extend their range further north to Alaska, their interior range is smaller and restricted to high elevations in the Rocky Mountains. Grand fir (*Abies grandis*

Dougl. ex D. Don Lindl.) has small coastal and interior ranges, while western white pine (*Pinus monticola* Dougl. ex D. Don) is more commonly located in mountainous areas.

Three species have a continuous range from the Pacific coast to the Rocky Mountains. These are lodgepole pine (*Pinus contorta* Dougl. ex Loud.), ponderosa pine (*Pinus ponderosa Dougl. ex Laws.*), and black cottonwood (*Populus trichocarpa* Torr. & Gray). Lodgepole pine possesses one of the widest species' ranges in western North America. It grows throughout the Rocky Mountain and Pacific coast regions, ranging from Yukon territory and the Pacific coast of Alaska to about latitude 31° N in California and Colorado. Ponderosa pine extends from southern Canada into Mexico and spreads towards the Pacific coast. The main range of black cottonwood extends along coastal Alaska through the forested areas of Washington and Oregon to northern California. This species is also found inland, generally in the Rocky Mountains of British Columbia, western Alberta, and northern Idaho.

### 2.5. Soil Properties

Slope was extracted from a Digital Elevation Model (DEM) of North America. The slope was classified as low (<4%), moderate (4%≥ to <16%), or steep (≥16%) according to the landform slope class developed by the government of Canada [46]. Slope was used as a proxy to estimate soil moisture because an increase in slope percentage implies increased drainage [47]. The rate of organic matter carbon decomposition (C/N) was extracted using a raster provided by the Food Agriculture Organization (FAO) world map with $5 \times 5$ arc minutes (FAO soils portal: https://www.fao.org/soils-portal/data-hub/soil-maps-and-databases/harmonized-world-soil-database-v12/en/, accessed on 2 April 2022).

### 2.6. Climate Variables

Climate data were obtained from the National Oceanic and Atmospheric Administration (NOAA: https://www.ncdc.noaa.gov/paleo-search/study/24230, accessed on 2 April 2022). The reconstructed anomaly of annual (AN_$T_a$) and summer mean temperatures (June-August; JJA_$T_a$) represents deviations from the annual mean (1961–1990) for northern hemisphere surface temperatures over a period of 998 years (1000–1998) and 1400 years (600–2002) for annual and summer mean temperatures, respectively, derived from the global climate proxy network [48,49]. Then, we extracted the real annual (AN_T) and summer mean temperatures (June-August; JJA_T) for each plot using ClimateWNA, a climate simulation software at $0.5° \times 0.5°$ resolution [50]. This software extrapolates climate variables from the nearest weather stations for the period 1901–2018. For the period before 1901, we previously used this software to extract the annual and summer mean temperatures for the period 1961–1990 corresponding to the base of the anomaly. Then, we were able to estimate the real temperature for each plot by adding the anomaly and the real mean temperatures for the period between 1961 and 1990. We also extracted annual and June-August Palmer Drought Severity Indexes (PDSI; [51]) for global surface temperatures over a period between 0 and 2000 years derived from the global climate proxy network gridded at ~$2° \times 2°$ resolution [52]. PDSI uses annual (PDSI_AN) and June-August (PDSI_JJA) rainfall levels to obtain a measure of drought to estimate the soil water budget for the period 0–2000 years. PDSI shows long-term dry (negative values) and wet (positive values) weather conditions. All plots in the same grid have the same PDSI. Averages that are greater than 0 represent a warmer/moister climate.

For each plot, these data were averaged for the period of tree growth under consideration (i.e., from the year of establishment to the year it reached a breast-height age of 50 years). Then, we related the establishment date to the averaged climate values.

### 2.7. Statistical Analyses

To answer the hypotheses, we first related site index, which is a proxy of the tree height growth, with the climate variables (AN_T, JJA_T, PDSI_AN, PDSI_JJA) and establishment date (ED) for each species using a Pearson correlation. Then, we built global and species-specific models: the global model allowed us to examine whether there was a general

trend in total tree height growth linked to climate change by relating the site index of all of the species to the environmental variables, such as soil conditions (slope and C/N), climate (AN_T and PDSI_JJA), the establishment date (ED), species' traits, and range groups. The variance components were estimated using the method of restricted maximum likelihood (REML). Since AN_T and JJA_T, as well as between PDSI_AN and PDSI_JJA, were highly correlated (multi-collinearity; Table S2), we selected only AN_T and PDSI_AN; this was because AN_T had a higher correlation value with the total species SI as compared to JJA_T, and PDSI_AN had a higher correlation frequency with site index compared to PDSI_JJA. Species were classified into different range groups based on their geographic area (Figure 1); species' traits used in the analysis were shade tolerance and leaf form (Table S1). A linear mixed model was used because species are included within their traits and range groups. We also included the interactions between the establishment date and other independent variables (soil conditions, species' traits, and groups) to highlight the climate-change-induced impacts on the historical site index pattern at different soil conditions and for different species' traits and groups. Thus, the interaction between ED and other independent variables was used as the fixed effect. ED was chosen for the base of the interactions and used as a proxy for climate change (Figure 2). Indeed, ED represents a combination of temperatures and precipitation over a specific period of time (years). Trees established at different times experienced different climate conditions (a combination of temperatures and precipitation) during their growth. A positive or negative effect of temperature or precipitation does not necessarily coincide with an increase or decline in historical growth. In fact, it is relatively easier to highlight 20th-century increases in warmth (Figure 2), but it is more difficult to model the general trend of precipitation [53,54]. Thus, trees growing during different time periods experienced different scenarios combining temperatures and precipitation (climate change): temperature increases and variations in precipitation imply different climate conditions, such as drought gradients, optimal conditions (no heat or water limiting), and different overflow gradients (an increase in temperature is not compensated for by higher precipitation levels) [55]. In addition, this method was previously used by Messaoud and Chen [30], who studied the historical height growth of black spruce (*Picea mariana*) and trembling aspen (*Populus tremuloïdes*) in British Columbia, Canada. In addition, species was used as a random effect to control for differences based on species; this made explicit the assumption that site index differs based on species.

For the species-specific model, we built a generalized linear model that included the same independent variables and interactions as in the global model. Since the study was at the species scale, we only excluded the species range groups, shade tolerances, leaf forms, and the interactions between these variables and the ED. However, geographic location (latitude, longitude, and elevation) and the interactions between location variables and ED were included, allowing us to investigate how the historical site index pattern changed geographically, especially in areas that experience heat and water limiting (high and low latitudes, western and eastern parts of the species' ranges, and high and low elevations). In addition, this model also assessed the site index pattern according to different soil conditions. Prior to analyses, a plot of the residuals was realized, and it was confirmed that the relationship was linear.

Species that have two distinct geographic ranges, such as Douglas-fir, grand fir, mountain hemlock, western hemlock, redcedar, and western white pine, were divided into coastal (Douglas-fir$_c$, grand fir$_c$, mountain hemlock$_c$, western hemlock$_c$, western redcedar$_c$, and western white pine$_c$) and interior (Douglas-fir$_i$, grand fir$_i$, mountain hemlock$_i$, western hemlock$_i$, western redcedar$_i$, and western white pine$_i$) subspecies. All analyses were performed with the R statistical software package [56] and the linear mixed model with the lme4 function in the lme4 package for R.

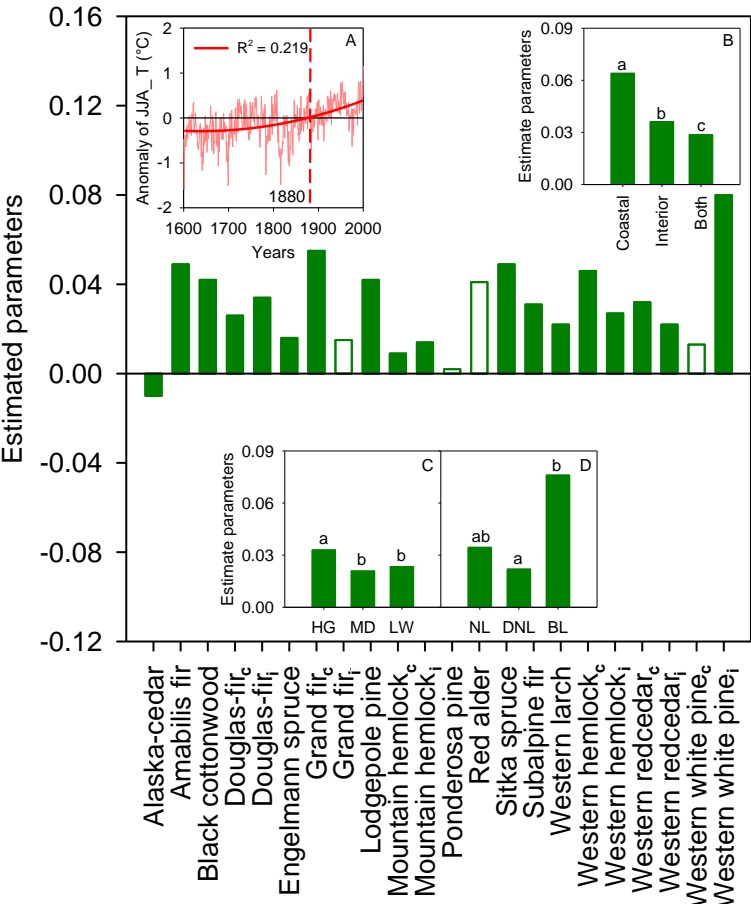

**Figure 2.** Estimated parameters of the simple regression trend between species site index (SI) and establishment date (ED). White bars indicate a non-significant relationship between SI and ED. The small panels indicate the evolution of the average June-August temperature (JJA_T) anomaly from the Northern Hemisphere mean temperature 1961–1990 in light red and the best-fitted regression line in dark red (**A**), the estimated parameters between SI and ED according to the species' range groups (**B**), shade tolerance ((**C**): HG = high, MD = medium, LW = low), and leaf form ((**D**) NL = needle leaf, DNL = deciduous needle leaf, and BL = broadleaf). The letter above each bar indicates significant (different letters) or no significant (same letters) differences between the groups, shade tolerances, and leaf forms.

### 2.8. Anomaly of Site Index

To highlight where and when the variation in SI occurs, we calculated the anomaly of site index, which is the variation from the mean value of the SI of the total study area for each species. Negative and positive anomalies indicate that a species' SI was below or above that mean. Then, a threshold of 20% of the variation of the mean SI was chosen to highlight where and when significantly higher or lower SI values occurred.

## 3. Results

### 3.1. Tree Site Index and Climate

Except for red alder, the site indexes (SIs) for all species correlated to at least one of the climate variables (AN_T, JJA_T, PDSI_AN, and PDSI_JJA) or to the establishment date (ED; Table 2). AN_T and JJA_T had the most frequent significant relationships with site index ($p = 0.508$ and $0.342$, respectively). The SI for coastal western white pine was not correlated with AN_T, and the SI for interior grand fir was not correlated with JJA_T, but AN_T had the highest correlation values globally compared to all other climate variables and ED ($p = 0.071$, $0.084$, and $0.235$ for PDSI_AN, PDSI_JJA, and ED, respectively).

**Table 2.** Pearson correlation values between site index, climate variables, and the establishment date (ED). In bold are significant levels (0.01 < α < 0.05, * 0.001 < α < 0.01, ** α < 0.001). In italics, 0.05 < α < 0.06. AN_T = annual mean temperature (°C), JJA_T = June-August mean temperature (°C), PDSI_AN = annual and PDSI_JJA = June-August Palmer Drought Severity Index. Subscript 'c' and 'i' indicate coastal and interior ranges, respectively.

| Species | AN_T (°C) | JJA_T (°C) | PDSI_AN | PDSI_JJA | ED |
|---|---|---|---|---|---|
| Alaska-cedar | **0.195 **** | **0.266 **** | **0.296 **** | **0.164 *** | **−0.255 **** |
| Amabilis fir | **0.554 **** | **0.552 **** | **−0.115 *** | −0.020 | **0.478 **** |
| Black cottonwood | **0.366 **** | **0.299 *** | **0.309 *** | **0.286 *** | **0.274 *** |
| Douglas-fir$_c$ | **−0.044 **** | **−0.239 **** | **0.183 **** | **0.143 **** | **0.182 **** |
| Douglas-fir$_i$ | **0.307 **** | **0.225 **** | **0.086 **** | **0.108 **** | **0.175 **** |
| Engelmann spruce | **0.459 **** | **0.452 **** | −0.026 | 0.006 | **0.157 **** |
| Grand fir$_c$ | 0.052 | 0.085 | 0.138 | 0.120 | **0.214 *** |
| Grand fir$_i$ | **0.172 **** | −0.011 | **0.084 *** | **0.090 *** | 0.040 |
| Lodgepole pine | **0.375 **** | **0.376 **** | **0.047 **** | **0.042 **** | **0.336 **** |
| Mountain hemlock$_c$ | **0.121 **** | **0.186 **** | **0.056** | **−0.142 **** | **0.254 **** |
| Mountain hemlock$_i$ | **0.401 **** | **0.401 **** | 0.118 | **0.164** | **0.323 **** |
| Ponderosa pine | **0.207 **** | **−0.145 **** | **0.099 **** | **0.094 **** | 0.008 |
| Red alder | 0.062 | −0.057 | 0.048 | 0.022 | 0.161 |
| Sitka spruce | **0.437 **** | **0.417 **** | −0.030 | **−0.105 *** | **0.433 **** |
| Subalpine fir | **0.394 **** | **0.336 **** | **−0.044 *** | $5 \times 10^{-4}$ | **0.306 **** |
| Western larch | **0.270 **** | **0.231 **** | 0.049 | 0.040 | **0.203 **** |
| Western hemlock$_c$ | **0.528 **** | **0.450 **** | −0.038 | 0.025 | **0.487 **** |
| Western hemlock$_i$ | **0.271 **** | **0.272 **** | −0.010 | 0.077 | **0.390 **** |
| Western redcedar$_c$ | **0.308 **** | **0.423 **** | **−0.191 **** | *−0.070* | **0.463 **** |
| Western redcedar$_i$ | **0.103 *** | **0.127 *** | **−0.104 *** | −0.073 | **0.286 **** |
| Western white pine$_c$ | 0.195 | *0.224* | 0.171 | 0.173 | 0.135 |
| Western white pine$_i$ | **0.210** | **0.255** | −0.029 | −0.029 | **0.313 *** |
| *Total* | **0.508 **** | **0.342 **** | **0.071 **** | **0.084 **** | **0.235 **** |

The SIs for seven species and subspecies were related to all climate variables or ED, while the SIs for coastal grand fir and coastal western white pine were only related to ED and JJA_T, respectively (Table 2). Among the climate variables, the SIs for Engelmann spruce, coastal and interior western hemlock, western larch, and interior western white pine were related only to temperature variables. However, no species had an SI related only to PDSI variables. Among the significant relationships, AN_T had a negative relationship only with the SI for coastal Douglas-fir, while JJA_T had a negative relationship with the SIs for coastal Douglas-fir and ponderosa pine. A negative effect of PDSI_AN on tree SI was observed for amabilis fir, subalpine fir, and the two subspecies of western redcedar, while a negative effect of PDSI_JJA occurred for coastal mountain hemlock, Sitka spruce, and coastal western redcedar. In addition, only the SI for Alaska yellow-cedar was negatively correlated with ED.

Since 1880, JJA_T was globally higher than the average and influenced the SIs of different species differently (Figure 2A). The estimated values represent the magnitude of the slope regression trend between SI and ED (Figure 2). Positive and negative values indicate an increase or decrease in SI through the study period, respectively. Except for Alaska yellow-cedar, all estimated values were positive, indicating an increase in SI within this period of time. The magnitude was greatest for interior western white pine and lowest for ponderosa pine.

### 3.2. Global Growth Patterns

The global model explained 49% of the variation in total species SI (Table 3). All explanatory variables had a significant influence on tree SI ($p < 0.05$). However, the significance level was strongest for AN_T and weakest for C/N.

**Table 3.** Summary of linear mixed model between site index (m), independent variables, and their interactions. C/N = organic matter decomposition rate, AN_T = mean annual temperature, PDSI_AN = annual Palmer Drought Severity Index.

| Fixed Effects | *F*-Value | *p*-Value | Random Effects | Variance | SD | Total Effects |
|---|---|---|---|---|---|---|
| *R² = 0.310* | | | | | | *R² = 0.490* |
| AN_T | 2125.83 | <0.001 | Species | 14.04 | 3.75 | |
| PDSI_AN | 141.04 | <0.001 | Residual | 40.11 | 6.33 | |
| Leaf form | 122.85 | <0.001 | | | | |
| ED | 95.76 | <0.001 | | | | |
| Groups | 24.27 | <0.001 | | | | |
| Shade tolerance | 19.02 | <0.001 | | | | |
| Slope | 7.75 | <0.001 | | | | |
| C/N | 4.07 | 0.044 | | | | |
| Groups*ED | 38.62 | <0.001 | | | | |
| Leaf form*ED | 16.35 | <0.001 | | | | |
| Shade tolerance*ED | 14.13 | <0.001 | | | | |
| Slope*ED | 7.52 | <0.001 | | | | |
| C/N*ED | 5.27 | 0.022 | | | | |

All interactions between ED and other explanatory variables had a significant effect on tree SI, with the strongest influence from the interaction between species' range groups and ED, and the weakest from the interaction between C/N and ED (Table 3). The increase was the greatest for the coastal species' range (group I) and the lowest for species with ranges covering both coastal and interior zones (group III; Figure 2B). High-shade-tolerant species demonstrated the greatest height increase (Figure 2C). The SI increased most for broadleaf species and was not significant for needle leaf species (Figure 2D). The greatest increase in SI was observed on low and steep slopes; additionally, it was magnified in areas with rich soil (Figure 3).

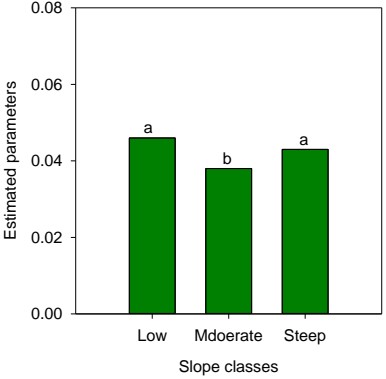

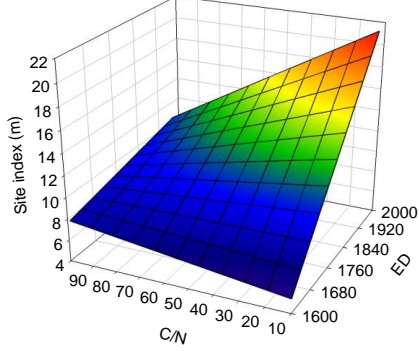

**Figure 3.** Relationship between SI and ED according to soil properties (slope classes and gradient of the soil decomposition rate; C/N). The letter above each bar indicates significant (different letters) or no significant (same letters) differences between the groups, shade tolerances, and leaf forms.

### 3.3. Species-Scale Growth Patterns

The species-specific model showed that the range of the variability of the SI explained by the model varied from 22.1% for Alaska yellow-cedar to 61.3% for ponderosa pine (Table 4). We note that this model explained more than half of the SI variability for 2 species (amabilis fir and ponderosa pine) and more than a third for 13 species and subspecies. Nevertheless, the SIs for six species were not significantly influenced by any interactions between variables.

**Table 4.** *F*-values of the general linear model between species site index and the independent variables. Only the interactions of the model are shown in the table. Lat. = latitude, Lon. = longitude, Ele. = elevation. In bold are significant levels (0.01 < α ≤ 0.05, * 0.001 < α < 0.01, ** α < 0.001). In italics, 0.05 < α < 0.06.

| Species | Lat. × ED | Lon. × ED | Ele. × ED | Slope × ED | C/N × ED | $R^2$ |
|---|---|---|---|---|---|---|
| Alaska-cedar | 1.14 | 1.61 | 0.05 | 0.83 | 0.57 | 0.221 |
| Amabilis fir | 2.47 | **10.23 *** | **34.04 ** ** | 1.11 | 0.35 | 0.606 |
| Black cottonwood | 2.35 | 0.27 | $3 \times 10^{-3}$ | 0.36 | **8.70 *** | 0.495 |
| Douglas-fir$_c$ | **103.15 ** ** | 3.20 | **5.84** | 0.52 | **10.80 *** | 0.335 |
| Douglas-fir$_i$ | **356.44 ** ** | **119.94 ** ** | **31.82 ** ** | **4.71 *** | **21.20 ** ** | 0.466 |
| Engelmann spruce | **29.42 ** ** | 1.19 | **29.17 ** ** | 2.44 | 0.06 | 0.378 |
| Grand fir$_c$ | 1.49 | 1.84 | 0.03 | 0.68 | 0.01 | 0.235 |
| Grand fir$_i$ | 1.79 | *3.66* | 2.37 | **3.91** | 0.13 | 0.449 |
| Lodgepole pine | **204.74 ** ** | **104.44 ** ** | 3.23 | **10.14 ** ** | **8.23 *** | 0.403 |
| Mountain hemlock$_c$ | **30.85 ** ** | **21.26 ** ** | **19.84 ** ** | 3.61 | 1.24 | 0.355 |
| Mountain hemlock$_i$ | 0.61 | 0.41 | 0.87 | 0.46 | 0.19 | 0.239 |
| Ponderosa pine | **26.61 ** ** | **8.10 *** | **65.24 ** ** | 0.39 | **29.29 ** ** | 0.613 |
| Red alder | 0.64 | 0.40 | 2.15 | **3.13** | $4 \times 10^{-3}$ | 0.223 |
| Sitka spruce | **10.12 *** | 0.10 | **13.13 ** ** | 0.08 | 0.08 | 0.379 |
| Subalpine fir | **87.52 ** ** | **42.98 ** ** | 0.30 | 1.74 | 0.02 | 0.352 |
| Western larch | **25.81 ** ** | 1.12 | 0.27 | 1.52 | 1.36 | 0.243 |
| Western hemlock$_c$ | **8.12 *** | **13.83 ** ** | **133.63 ** ** | **5.48 *** | 0.41 | 0.446 |
| Western hemlock$_i$ | 1.64 | **6.89 *** | **8.67 *** | 1.05 | 0.35 | 0.267 |
| Western redcedar$_c$ | **32.41 ** ** | **4.48** | 2.59 | 0.04 | 0.37 | 0.352 |
| Western redcedar$_i$ | 0.03 | 0.02 | 1.60 | 1.40 | 0.06 | 0.242 |
| Western white pine$_c$ | 3.23 | 1.47 | 0.44 | 0.46 | 2.72 | 0.309 |
| Western white pine$_i$ | 0.02 | 0.62 | 0.47 | 0.05 | 0.60 | 0.317 |

The interaction between the ED and the soil properties demonstrated that the best SI occurred on low and moderate slopes for interior Douglas-fir and coastal mountain hemlock (Figure 4). The SIs for interior Grand fir, lodgepole pine, red alder, and coastal western hemlock did not significantly differ with slope classes, although higher SI values were observed on steep slopes for interior Grand fir and red alder. With the fertility gradient, the largest significant increases in SI under rich soil conditions occurred for black cottonwood, coastal and interior Douglas-fir, and lodgepole pine, whereas SI was greater in poor soil conditions for ponderosa pine (Figure 5). Only black cottonwood and interior Douglas-fir showed a decline in SI under poor soil conditions.

The SI–ED-induced pattern demonstrated that all species saw a greater increase at higher latitudes compared to lower latitudes except for Sitka spruce, coastal western hemlock, and coastal western redcedar, for which it was greater at lower latitudes (Figure 6). Only coastal western redcedar demonstrated a slow decline in SI at higher latitudes. In contrast, five species demonstrated a decline in SI at low latitudes. With longitudes, all significant interactions showed that the largest increase occurred moving westward, except for coastal western hemlock and redcedar, for which it increased moving eastward. All significant interactions between SI and ED at different elevations showed that all species demonstrated higher SIs at lower elevations. Further, a decline in SI occurred at higher elevations for coastal and interior Douglas-fir, coastal mountain hemlock, ponderosa pine, Sitka spruce, and coastal western hemlock.

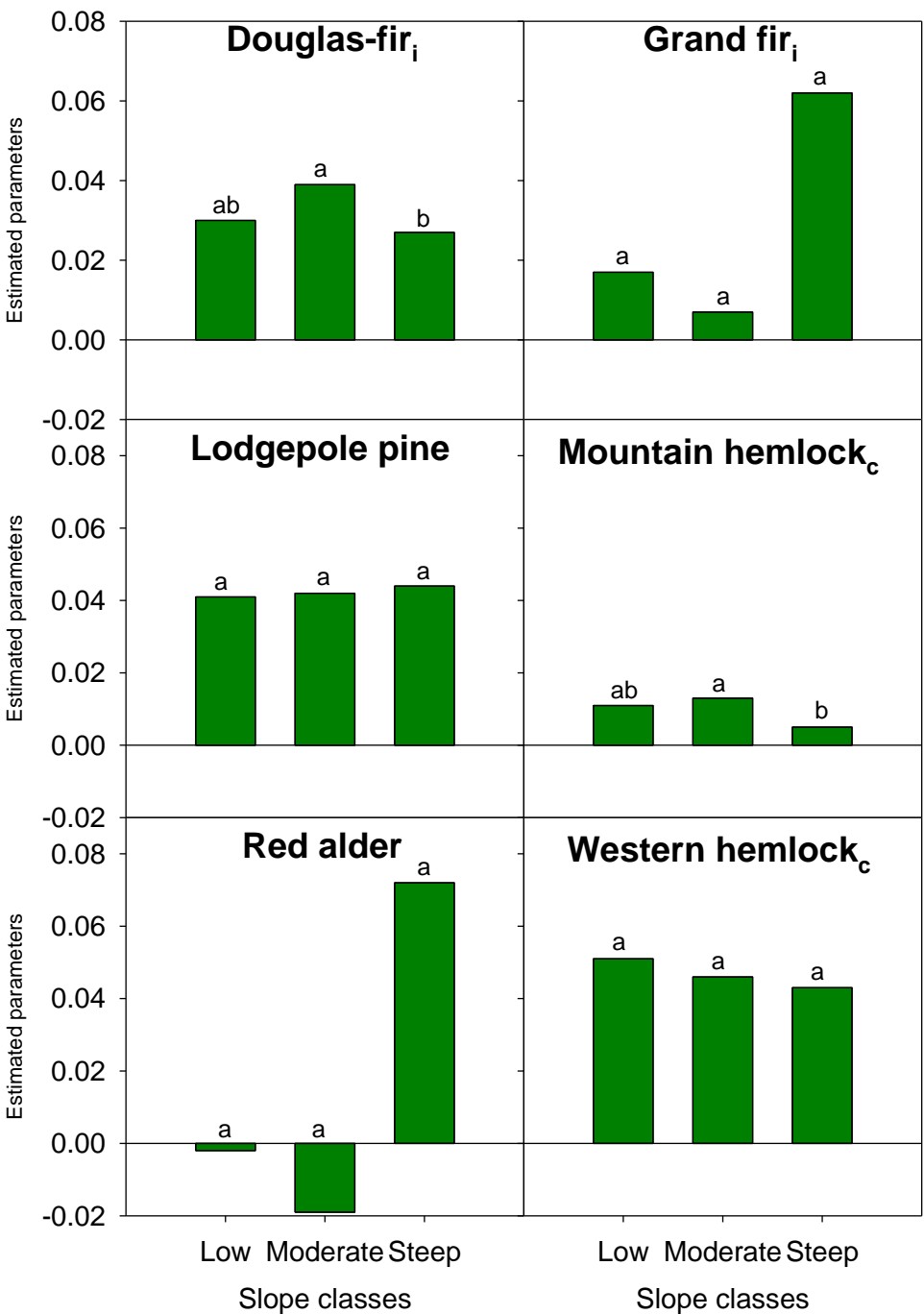

**Figure 4.** Estimated parameters of simple regression between species SI and ED for different slope classes. The letter above each bar indicates significant (different letters) or no significant (same letters) differences between the groups, shade tolerances, and leaf forms.

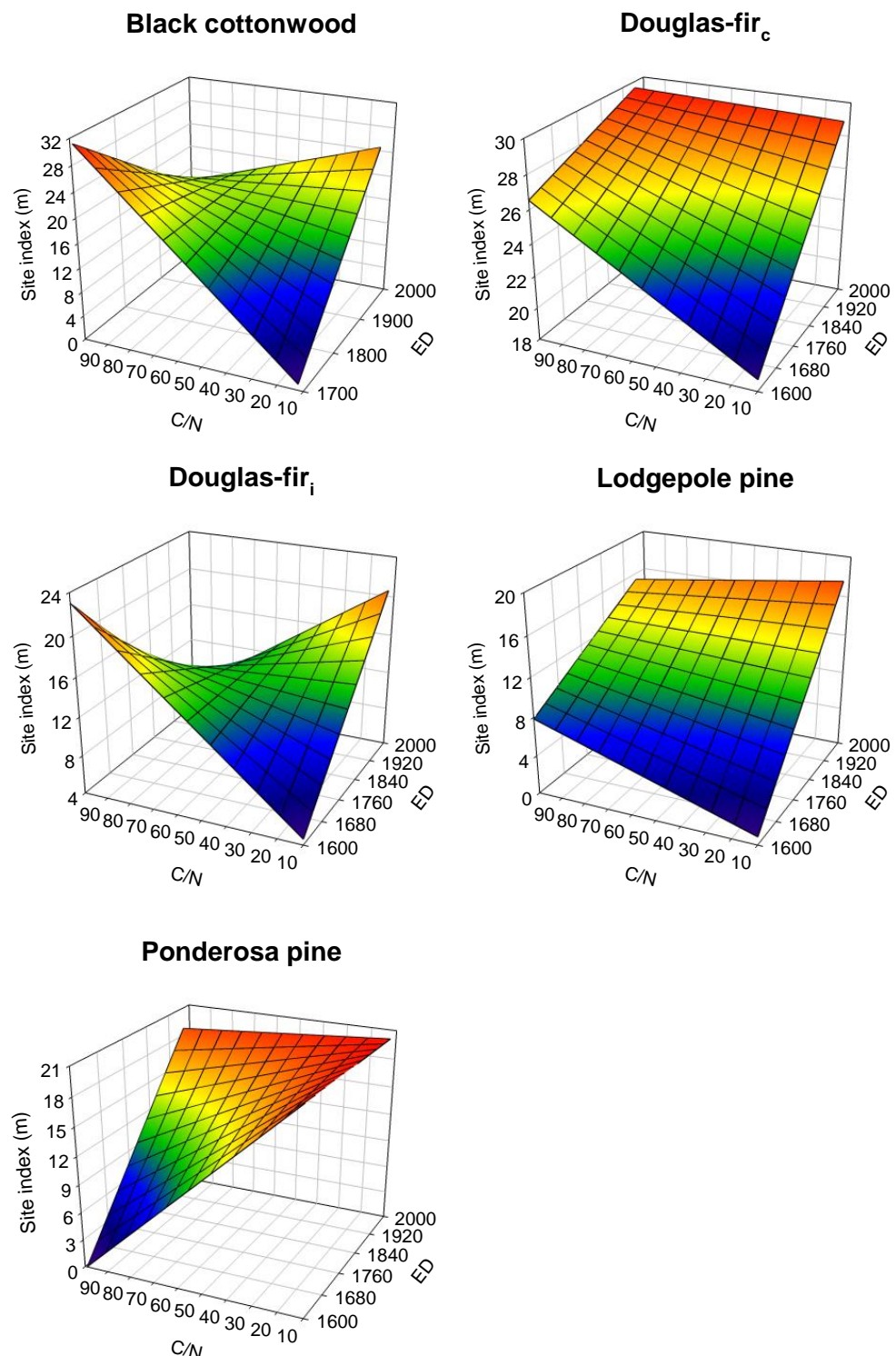

**Figure 5.** Relationship between species SI and ED according to the gradient of the soil decomposition rate; C/N.

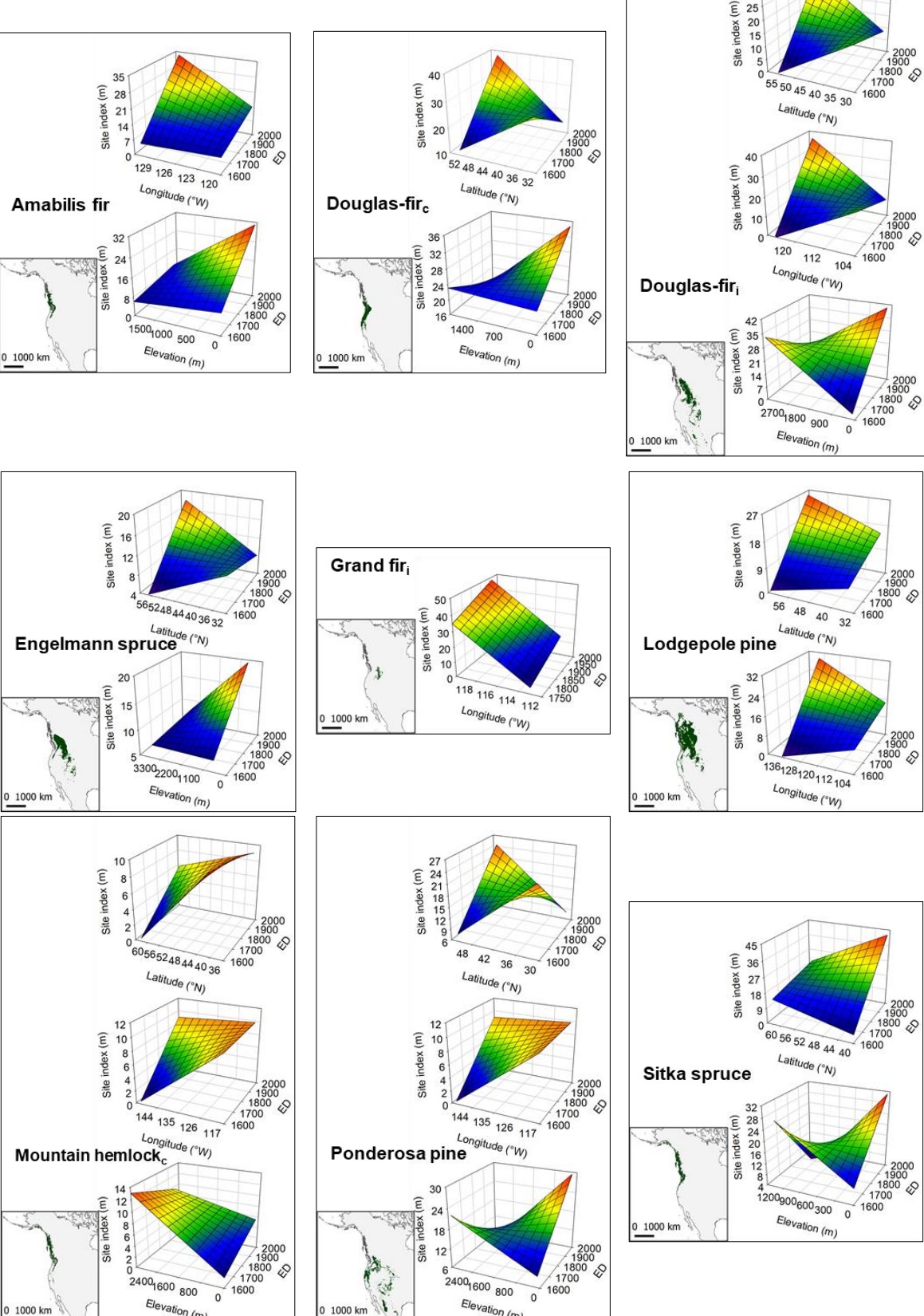

**Figure 6.** *Cont.*

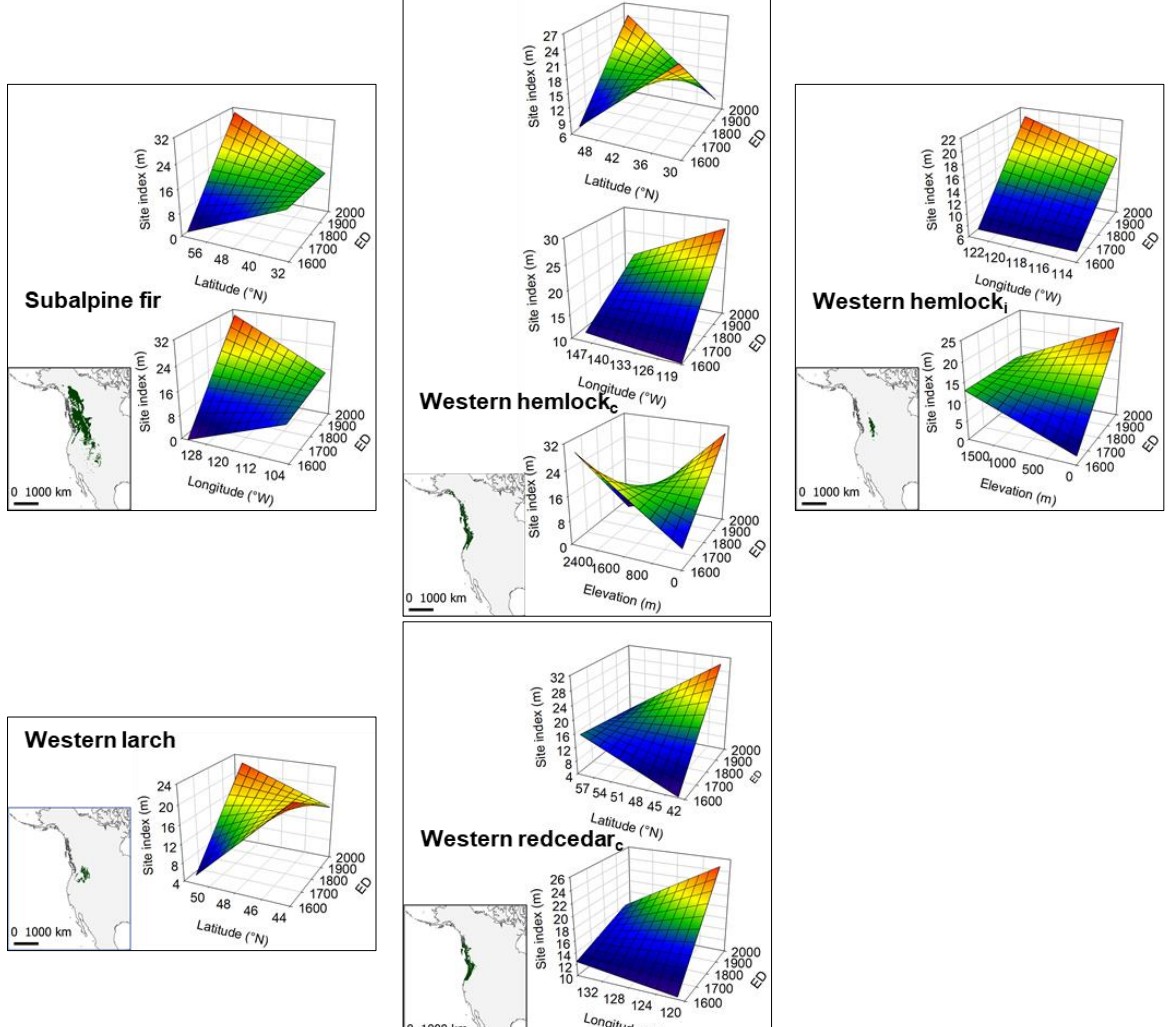

**Figure 6.** Relationship between species SI and ED for different geographic locations. Inset maps indicate species' geographic ranges.

*3.4. Anomaly of Species Site Index and Climate Change*

The anomaly of SI, representing the deviation from the mean of the total species SI of the study area (see Section 2.7 of Materials and Methods), differed with species, establishment date, and geographic location (Figure 7). During the warm period (1880 and afterward), Alaska yellow-cedar was the only species that had the highest and the lowest frequency of plots where the anomaly was below and above the threshold (42.86 and 11.90%, respectively). The plots of Alaska yellow-cedar with a site index anomaly below the threshold occurred mostly after the middle of the 18th century and at the northernmost portion of the species' range. Most species that had plots with a positive site index anomaly greatly overwhelmed the threshold, especially during the warm period. However, interior Douglas-fir, Engelmann spruce, interior grand fir, and ponderosa pine had slightly more plots with negative values (Figure 7). In addition, plots with negative values, i.e., values below the threshold, occurred mostly in the southernmost extent of the species' ranges, while plots with positive values, i.e., above the threshold, occurred mostly in the center or in the northernmost extent of the species' ranges. Conversely, plots with positive values (above the threshold) occurred mostly in the southernmost extent of the species' ranges for Sitka spruce, coastal western hemlock, and redcedar; to a lesser extent, the same was true for coastal mountain hemlock.

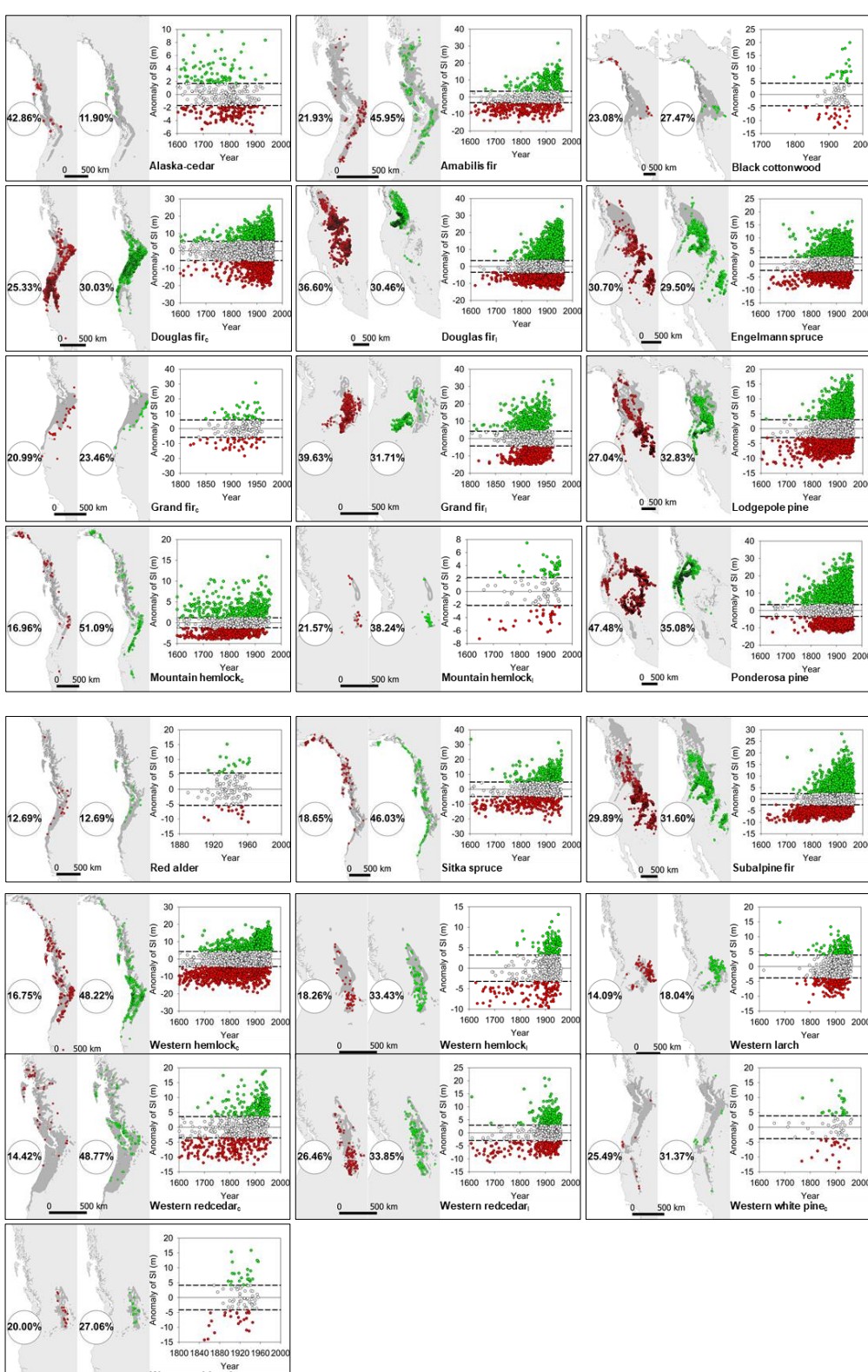

**Figure 7.** Graphs show the evolution of the anomaly of species SI on each plot according to ED. The anomaly represents the deviation from the mean SI of each species within the whole study area (Table S3). The long-dashed lines indicate the threshold of 20% of the mean SI value. Red and green circles indicate values below and above this threshold, respectively. Maps show the distribution of the plots where these values occurred, and the percentages indicate the proportion of these plots among those established during the warm period (1880 and afterward; see Figure 2a).

## 4. Discussion

### 4.1. Relationship between Height Growth and Temporal Variables

Except for Alaska yellow-cedar, all study species that were established during the 20th century showed higher SI values than those established before that period. Increases in temperature and precipitation of 0.8 °C and 14%, respectively, during the 20th century in the Pacific Northwest of North America [57–59] likely contributed to the height growth increase (Figure 7). An explanation for this trend is that longer growing seasons since 1880 (Figure 3) may have promoted earlier plant growth [60]. Temperature variables and ED were better for predicting height growth at both the global and species levels for species in western North America compared to the two PDSI variables, similar to previous studies [33,61,62]. Further, Hayes et al. (1999) noted that PDSI might be less suitable for predicting growth in mountainous areas with frequent climatic extremes and more variation in precipitation and soil moisture.

As predicted, increased temperatures generally had a positive effect on tree height growth when water availability was not a limiting factor [62–65]. In non-water limiting areas, heat promotes photosynthesis and carbohydrate allocation to the stem [14]. The exception was height growth for coastal Douglas-fir and ponderosa pine, which were negatively correlated to both temperature variables, especially to JJA_T (Table 2); this may be explained by water limitations. Ponderosa pine is found in dry climates and on sandy soils [43,44]. Consequently, an increase in JJA_T may put trees in a higher water stress situation [66]. Coastal Douglas-fir is found mostly in regions characterized by a Mediterranean climate, with a high amount of winter precipitation but hot and dry summers [67]. Thus, an increase in JJA_T may exacerbate evaporation and water stress sensitivity during the summer growing season [68]. Nevertheless, the absence of height growth decline during the 20th century indicates that these two species globally overcame their growth limitations, perhaps by using water that condensates from fog, which, in turn, arises from contact between the warm air above the coast and the cooler water of the Pacific, especially for coastal Douglas-fir (Figure 2; [69]).

The negative effect of PDSI_AN and PDSI_JJA on tree height growth for four and three species and subspecies, respectively, appears to have two major causes. First, soil water surplus may decrease root activity due to asphyxia and photosynthetic activity; this would reduce growth [14,70,71]. Second, cooler summer temperatures and snowpack accumulation in mountainous areas can negatively affect tree height growth. Indeed, Appleton and St. George [72] found that the growth of mountain hemlock in the central Pacific coast was negatively correlated to cool-season precipitation, corroborating our findings. The increase in PDSI_AN may also be related to an increase in snow cover during the winter at higher elevations. A deep snowpack, not measured in the study, decreases soil temperatures and thus the length of the growing season at the local scale, and a long-lasting snowpack serves to limit the growth of high interior elevation species and subspecies [73]. Kajimoto et al. [74] found that heavy snow accumulation was responsible for tree damage and thus reduced growth.

Only Alaska yellow-cedar demonstrated a negative correlation with ED and had the largest decline in height growth during the 20th century (Table 2; Figure 2). This was unexpected but perhaps not surprising and may be explained by other factors not included in our study. Beier et al. [75] also found a decline in radial growth for Alaska yellow-cedar during the 20th century, likely related to warmer winters resulting in less snowpack and therefore less protection of surface roots from freeze-thaw events, leading to root injury [76]. Schaberg et al. [77] found that the roots of Alaska yellow-cedar were more vulnerable to freezing compared to Sitka spruce, western and mountain hemlock, and western redcedar.

### 4.2. Global Height Growth Patterns

The global model explained 49% of the total variability in species' height growth (Table 3), corroborating the tenet that tree height growth is strongly influenced by climate, soil conditions, and species characteristics [30,78]. However, the historical height growth

relationship with ED differed in magnitude according to different geographic locations, species characteristics, and soil properties; this was in line with our expectations. This confirms that historical tree height growth patterns were not uniform for all studied species or for the whole study area, as mentioned in the work of Hararuk [29]. However, our findings corroborated the findings of Messaoud and Chen [30], Sullivan et al. [31], and Cahoon et al. [17], suggesting that the trends found here represent the general trend in western North America.

Coastal species (group I) demonstrated a higher increase in height growth with ED compared to other species groups and benefited more from climate change in the 20th century (Figure 2). Species in group II had the second-highest increase in height growth; these species are sensitive to warmth because they are located mostly in the elevated Rocky Mountains and thus are more temperature-driven, whereas the coastal Pacific region is a temperate rainforest and benefits from a mild temperature with a high water availability, i.e., frequent rain and/or fog [63,79,80]. This temperate rainforest is the largest and one of the most productive in the world [81], which may explain the fact that the greatest benefit of a warming climate was observed for the species in this area. Moreover, some species, such as Sitka spruce, require a large amount of nutrients, such as phosphorus, that are provided by ocean spray and brackish water [44]. Coops et al. [82] argued that the high winds and salt spray typical of coastal regions might be correlated with the environmental characteristics associated with the performance of Sitka spruce.

The hypothesis that fast-growing species benefit more from climate change than slower-growing species [83,84] is not corroborated by our study since the increase in height growth with ED was higher for high-shade-tolerant (slowest-growing) species compared to mid- and low-shade-tolerant (faster-growing) species. A similar increase in height growth between mid- and low-shade-tolerant species appears to have two explanations. First, species located in the coastal temperate forest benefitted the most from recent climate change and the particular fertilization of the ocean mentioned above; this masks the effect of shade tolerance. Second, some mid-shade-tolerant species, such as interior western white pine, exhibited the greatest increase in height growth with ED.

The other hypothesis, that deciduous broadleaf species exhibit a greater climate growth response than coniferous species [15], is supported by our findings. Trees with high leaf photosynthetic capacity, i.e., broadleaf trees, can achieve their growth and acquire resources rapidly in order to maintain high growth rates under environmental conditions such as those created by recent climate change [85], while needle leaf species allocate their resources not only for their growth but also for leaf protection during the winter period.

Tree height growth was equivalent on low and steep slopes and greater than on moderate slopes values, while tree height growth demonstrated the highest values on high-soil-fertility sites (Figures 3 and 4), which are considered to have good soil quality for growth (mesic soil; [34,86]). Leonelli [87] found that historical increases in height growth for trembling aspen in British Columbia were greater on rich soil.

### 4.3. Species-Specific Growth Patterns

We found increased height growth for ponderosa pine in conditions of low soil fertility, contrary to the norm (Figure 5). This could be explained by these sites having higher organic matter content, which is known to result in higher water-holding capacity [88,89]; this is crucial for species' growth in dry climates. Furthermore, Lévesque et al. [34] found that, in contrast with other tree species growing in central Europe, recent tree growth for Scots pine and oak species was relatively good on poor and dry soil. They explained that these two dry-climate-adapted species have a competitive advantage over water- and nutrient-demanding species in poor and dry soils.

Several studies have reported different tree growth patterns according to geographic location, with increases at higher latitudes and elevations and a decline at lower latitudes and elevations due to drought conditions [79]; these results were only corroborated by our findings for high latitudes (Figure 6). At higher latitudes, an increase in temperature for

more limiting environments had a stronger and more beneficial effect on tree height growth than at low latitudes [90]. Higher increases in growth for most species at the westernmost extent of their ranges and at lower elevations indicated that locations close to the Pacific coast benefited from a longer growing season and increased precipitation. A decline in height growth observed for southward and eastward sites demonstrated the possible effects of drought occurrence at the eastern edge of the species' ranges because an increase in temperature brought trees under water deficit stress [91]. In fact, an obvious southward decline in height growth response for coastal Douglas-fir corresponded to the coastal areas of California characterized by Mediterranean climate conditions known for summer droughts [92] and also to the drier conditions that occurred in the interior southwestern U.S. for other species. Indeed, western North America is known for its peculiarity, in that high latitudes correspond mostly to low elevations and vice versa (Table S2; [93]). As well, an unexpected decline in height growth for many tree species at higher elevations appeared to be related to increases in cloudiness or water stress in interior dry conditions, impeding the positive effects of recent increases in temperature [23].

In contrast, we observed slower growth for a few coastal species and a decline for coastal western redcedar at the leading range and upward, corresponding to the coastal and mountainous areas of the Pacific region, respectively, where they reach their northernmost limit [45]. Further, increased precipitation in the 20th century likely increased cloud cover, which for regions corresponding to limiting temperatures can reduce summer temperatures and photosynthesis [94].

### 4.4. Spatio-Temporal Variation in Species' Tree Growth

The anomaly of site index gives additional information on the amplitude of the variation in species height growth. Most species increased their height growth by at least 20% (the threshold) of the species' mean SI on many plots, especially during the warm period (1880 and afterward; Figure 7), supporting the previous findings that a warmer climate positively affects species' height growth. Only five species had a higher percentage of plots with anomaly values below the threshold, with the highest values for Alaska yellow-cedar (42.86%). For this species, the trend indicated that few plots had a positive anomaly of SI, supporting the conclusion that there was a negative effect of warming on these species' height growth. The plots with negative values below the threshold were mostly located at the northernmost extent of the species' range, supporting the decline of Alaska yellow-cedar in the southernmost region of Alaska [75,76]. For the remaining four species, we found many plots with high positive anomalies, which compensated for the higher proportion of plots with negative anomalies (Figure 7).

Geographically, most plots with height growth anomalies over the threshold were located at the northernmost extent of the species range, supporting the previous findings. In addition, Sitka spruce, coastal western hemlock, and redcedar magnified their height growth mostly at their southernmost ranges, globally corroborating our previous findings. In contrast, only a few species show an opposite trend; interior grand fir and ponderosa pine, which had their height growth anomaly over the threshold throughout the west toward the Pacific area, avoided the drought conditions in the interior part of the study area [95–97]. A negative height growth anomaly below the threshold was also found for ponderosa pine in the northernmost extent of the species' range, corresponding to the southeastern part of British Columbia. This area represents the ponderosa pine zone known as the driest of the forested zones in the province [98].

### 4.5. Growth Patterns with Distinct Geographic Ranges

Comparing between coastal and interior ranges, species demonstrated different growth patterns and sensitivities to climate change (Figures 2 and 7). Greater growth was observed in coastal than in interior ranges (grand fir, western hemlock, western red-cedar), while it was the opposite for the coastal ranges (Douglas-fir, mountain hemlock, and western white pine). As mentioned above, coastal and interior Douglas-fir benefited

differently from climate change due to the differences in summer climate conditions between coastal and interior ranges, which is also the case for western white pine, although the magnitudes for the two species differed. However, the same trend was found for mountain hemlock, and the explanation is related to an increase in cloudiness (mentioned above) during the growing season in more temperature-limiting areas; this was more pronounced in the coastal than in the interior ranges.

In contrast, coastal grand fir occurs in non-water limiting zones (no significant correlation with PDSI_AN or PDSI_JJA), which magnifies its height growth with climate change compared to interior grand fir, for which the water requirement appears to be higher [44]. As well, coastal western hemlock and redcedar showed greater height growth compared to the interior varieties of these species, as was found for grand fir. Furthermore, we found that interior species were globally less sensitive than coastal ones. This could be related to possible non-climatic influences on these species. In fact, there is evidence that the recent geographic distribution of these two species is still not in equilibrium with the climate [99,100]. We argue that, in the interior ranges, hemlock and redcedar are not as controlled by climate as compared to these species in their coastal ranges. For instance, Western hemlock became common in its interior range only 2000–3500 years ago, compared with over 9000 years ago in the coastal region. The lack of a link between species' performance and climate conditions was pointed out in other parts of North America, such as jack pine [101] or lodgepole pine in the Yukon [102], supporting the hypothesis that Western hemlock and redcedar have not completely expanded into their potential habitat.

## 5. Conclusions

In western North America, the relationship between historical tree height growth and climate change demonstrated that temperatures were the best predictors of growth dynamics. 20th-century warmth was beneficial for all study species, except for Alaska yellow-cedar, corroborating our first hypothesis.

Furthermore, coastal, high-shade-tolerant, and deciduous broadleaf species and subspecies benefited the most from 20th-century warmth, while species with ranges spanning coastal and interior regions, middle-shade-tolerant species and subspecies, and deciduous needle leaf species, represented by western larch, benefited the least. This demonstrates that different growth patterns are found with different growth habits, leaf forms, and spatial environments, in line with our second and third hypotheses.

In addition, only for red alder was historical height growth related neither to the independent variables nor to their interactions. Furthermore, red alder was also the only species that had the same proportion of plots with height growth below and above 20% of the mean SI. Thus, our results demonstrate the complexity of the climate-change-induced growth response at the subcontinental scale, where both global and species-specific trends in height growth occurred simultaneously and differed geographically and according to different soil qualities. Climate-change-induced growth patterns could be used as a proxy to highlight potential interspecific growth competition and thus detect possible spatio-temporal shifts in species distribution, especially at the species' leading and tailing ranges. However, performance under future climate change scenarios could be more complex than can be anticipated by simulated predictions because of the complexity of tree responses to climate change, where species' ecological amplitudes and traits play a stronger role at a larger scale.

**Supplementary Materials:** The following supporting information can be downloaded at: https://www.mdpi.com/article/10.3390/f13050738/s1, Figure S1: Major ecoregions of North America. Table S1: Characteristics of the study species; Table S2: Correlation coefficient between the continuous explanatory variables; Table S3: Average of the whole plots SI with standard deviation for each species.

**Author Contributions:** Y.M. conceived the original study idea, designed the methodology, and led the analysis of the data and the writing of the manuscript. A.R. revised the manuscript and performed the English editing. J.A.G. revised the statistical part of the manuscript and participated in improving

the two statistical models. N.M.T. and A.H. contributed critically to the drafts and gave final approval for publication. All authors have read and agreed to the published version of the manuscript.

**Funding:** This research received no external funding.

**Institutional Review Board Statement:** Not applicable.

**Informed Consent Statement:** Not applicable.

**Data Availability Statement:** The data that support the findings of this study are available for the U.S. (https://www.fia.fs.fed.us/, accessed on 2 April 2022), but those from Canada are under a license agreement and are thus not publicly available. The data are, however, available upon reasonable request and with permission from the appropriate Canadian national or provincial government.

**Acknowledgments:** The authors acknowledge Shirley Mah and Gordon Nigh for providing the Site Index Biogeoclimatic Ecosystem Classification data (SIBEC). Many thanks to Rene Delong, Tom Malone, Marin Palmer, and James Menlove from Forest Inventory Analyst in the U.S. and Kirk Price from Forest Management Branch in Yukon. Special thanks to Han Chen for his substantial help during the earlier stage of the project, Michael Ter-Mikaelian for their substantial comments and suggestions, Faouzi Messaoud for his help to extract the big data with a special software and Shukry Messaoud for his final English revision. This research did not receive any specific grant from funding agencies in the public, commercial, or not-for-profit sectors.

**Conflicts of Interest:** The authors declare no conflict of interest.

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
