# Peer review of "The Historical Complexity of Tree Height Growth Dynamic Associated with Climate Change in Western North America"

_forests, doi:10.3390/f13050738_

Round 1
Reviewer 1 Report
The authors present an extensive analyses on the effect of temperature increase over height growth of sixteen tree species across western North America. I found the paper particularly interesting because most studies on climate change effects over tree growth, have analyzed tree ring increment. They concluded that most tree species analyzed were benefitted by higher temperatures in the 20th century.
Generally, this paper has a clearly articulated message, and by directly addressing a significant question on the effect of climate change, it will be of interest to a broad audience. However, since it is a bit long by the number of analyzed species, some parts of the paper need a careful writing reviewing, for better understanding of the potential readers.
Particularly important is the definition of height increment (height growth increase) derived from the Site Index (SI) determination (“as a proxy of tree height growth“… line 240-241). I guess the dataset come from plots measured only once, that not necessarily report a “true” height increment overtime. In Table 1, a range of SI in presented in the last column. The lowest value is 1.82 m for Subalpine fir. Does in mean that a tree of such height has at least 50 years? Of course, this is totally possible, however, coring a tree of such dimensions for age determination would be “very difficult”. Some other species SI ae reported to be less than 3 m at is lower interval.
General:
The figures would require some editing process before I would consider them to be of acceptable publication quality. For example, in some cases the labels and legends are too small and difficult to read. In Figure 2, c for coastal and i for interior are not explained, and the letter that identifies the panel is confusing with those indicating the statistical differences.
What the difference between “species and populations” is? In serval sentences in results and discussion sections is mentioned this expression (such as in loine 317, 355, …).
Line 154-156. “… dominant and co-dominant (largest trees) living trees for each species were aged by counting tree rings from an increment core sample extracted at the breast height of a tree where the diameter is measured at root collar for each species”. Is it DBH or is it diameter at the root collar?
Line 166-167. The link provided does not take you to the referred software.
Line 212-213. The link does not work.
Line 286-287. Review the sentence
Line 289. “…shade tolerance leaf form”. A comma is necessary.
Line 514. “low and steep slopes”… is this correct? Where the moderate slope is? I would expect an intermediate behavior in this latter slope condition.
Table S1. I guess “Tolerance” refers not only to shade but also to other stresses indicates in the second line. If it is the case, “Tolerance” should be centered.
Table S3. Is not 5.97 m as mean SI for mountain hemlockc too low?
Reviewer 2 Report
This is a timely and interesting study. I do only have some minor questions related with methodological aspects
Line 141: Table 1 contains 48440 plots. Where does this difference come from?
Line 143: Why did you select stands established after a wildfire?
Line 150: Here you mention plantations but in line143 you state that stands were “naturally” established. Similarly, you mention that stands were selected “unmanaged, and without visible damage or disturbance.”, but in the stand conditions recorded you mention “visible natural or anthropic disturbance, treatment, plantation or natural establishment”. Please clarify
Line 152. Were both monospecific and mixed stands included? If mixedced stands are considered, what is the potential effect for the calculations related with the site index?
Line 154: What is the distinction between a dominant and a co-dominant tree?
Line 214. I am sorry but, while I understand how climate variables were obtained I do not.
Line 400. The anomaly in the SI is a key variable to define the differences in species responses. I suggest defining it in the Material and Methods section.
Reviewer 3 Report
In the "Conclusion" section, it could be briefly stated separately for each of the three hypotheses, whether it was confirmed or not. At the same time, I recommend moving and/or omit the mention of hypotheses from the "Results" section.
